# In Vitro Accuracy of Digital and Conventional Impressions for Full-Arch Implant-Supported Prostheses

**DOI:** 10.3390/jcm11030594

**Published:** 2022-01-25

**Authors:** Rani D’haese, Tom Vrombaut, Herman Roeykens, Stefan Vandeweghe

**Affiliations:** Oral Health Sciences, Faculty of Medicine and Health Sciences, Ghent University, 9000 Ghent, Belgium; Rani.DHaese@ugent.be (R.D.); Tom.Vrombaut@ugent.be (T.V.); Herman.Roeykens@UGent.be (H.R.)

**Keywords:** intraoral scanning, accuracy, full-arch implant impression, digital impression, edentulous

## Abstract

The aim of this study was to evaluate the accuracy of full-arch digital impressions when compared to conventional impressions, when performed on the abutment or implant level. Methods: One resin cast with six implants and another cast with six abutments were scanned with Primescan v5.1 (PS51), Primescan v5.2 (PS52), Trios 3 (T3), and Trios 4 (T4). Additionally, conventional impressions (A) were made, poured in gypsum, and digitized using a lab scanner (IScan D104i). A coordinate machine (Atos, GOM, Braunschweig, Germany) was used to generate the reference scan of both casts. For all scans, the position of the implants was calculated and matched with the reference scan. Angular and coronal measurements per implant were considered for trueness and precision. Results: For the implant-level model, PS52 performed significantly better in terms of trueness and precision compared to all other impressions, except for the angular trueness of A (*p* = 0.072) and the coronal trueness of PS51 (*p* = 1.000). For the abutment-level model, PS52 also performed significantly better than all other impressions, except for the coronal trueness and precision of A (*p* = 1.000). Conclusions: Digital impressions for full-arch implant supported prostheses can be as accurate as conventional impressions, depending on the intra-oral scanner and software. Overall, abutment level impressions were more accurate compared to implant level impressions.

## 1. Introduction

The introduction of Cad-Cam technology has made it possible to scan and create a 3D digital image of a tooth preparation or implant, which can be used to design and fabricate a restoration. Over the years, various applications of this data acquisition system were developed in different aspects of dentistry [1]. It is routinely used for the fabrication of crown and bridges, but can also be used to plan surgeries such as cleft palate or other jaw surgeries virtually.

In prosthodontics, intra-oral scanning has simplified the impression procedure by reducing the number of production steps. This improves the precision, reduces the treatment time, and finally leads to a better fit of the restoration compared to conventional impressions [2,3,4]. However, the accuracy is inversely proportional to the size of the area to be scanned [5]. Therefore, intra-oral scanners are a valid alternative to conventional impressions for partial arches, but still challenging for complete-arch impressions, although some devices have shown comparable results [6].

Compared to tooth preparations, dental implants are more challenging to take an accurate impression. Errors caused by the displacement of different components can accumulate rapidly when multiple implants are involved [7]. To date, the literature has validated the use of IOS for short span implant-supported restorations, but there has been some reluctance for completely edentulous implant cases due to inconsistent results [8,9,10,11]. After all, the lack of landmarks in the mucosa complicates the stitching of the different images taken by the scanner [1,11,12].

Some recent studies have demonstrated acceptable accuracy for complete-arch implant impressions [7,11,13]. However, many factors influence the accuracy of intra-oral scanning, which can explain the various findings and wide range reported in the literature. This variation might be caused by operator experience, scanning conditions and strategy as well as different types of scanbodies [11].

An accurate impression is crucial to achieve a passive fit of the prostheses. This is mandatory since osseointegrated implants are not able to compensate for inaccuracies due to the absence of a periodontal ligament [14]. Even small discrepancies can induce misfit, which generates tension and compromises the long-term success [10,15]. The use of screw retention in combination with stiff ceramic materials has even increased the demand for an accurate fit.

Wulfman et al. [9] states that a misfit of 150 µm is considered acceptable, as it does not induce clinical complications. A more recent study puts the threshold at 59–72 µm linear and 0.4° angular displacement between implants [16], which corresponds to a maximum lateral movement of 50 µm of the implant in bone [15]. However, different reviews have concluded that there is no consensus [10,17]. Di Fiore et al. [10] believe that the errors should be limited to 30–50 µm to avoid mechanical and biological complications. To maintain a passive fit, 30 µm should be the aim [18].

Accuracy is defined by “trueness” and “precision”, as described in the ISO 5725 standard. “Trueness” refers to the closeness of agreement between the arithmetic mean of a large number of test results and the true or accepted reference value. “Precision” refers to the closeness of agreement between test results.

The aim of this study was to evaluate the accuracy of full-arch digital impressions when compared to conventional impressions when performed on the abutment or implant level. The first null hypothesis states that there are no differences in accuracy between the open tray analogue impressions and the different digital impressions. The second null hypothesis states that no difference in accuracy can be found between the implant level or abutment level impression.

## 2. Materials and Methods

Two identical edentulous resin mastercasts were printed with a 3D printer (Asiga UV Max 3D printer, Sydney, Australia) and six implants (Astra Tech Implant EV 4.2S–13 mm, Dentsply Sirona Implants, Mölndal, Sweden) were placed in each cast, at the site of the lateral incisor, first premolar, and first molar. The implants were placed freehanded by an experienced surgeon and as parallel as possible. On one of the casts, six abutments (Uni Abutment EV, Dentsply Sirona Implants, Mölndal, Sweden) were connected to the implants and torqued at 25 Ncm.

Both mastercasts were scanned 10 times using Primescan (Dentsply Sirona, Bensheim, Germany) with software version 5.1 (PS51), Primescan with software version 5.2 (PS52), TRIOS 3 (T3), and TRIOS 4 (T4) (3Shape, Copenhagen, Denmark) without the scanbodies in situ. The scanners were calibrated according to the manufacturer’s instructions prior to use. Next, one-piece titanium scanbodies (IO FLO-S, Dentsply Sirona Implants, Hanau, Germany) with a sandblasted surface, were hand-tightened to the implants on one mastercast and to the abutments on the other mastercast, followed by a secondary scan to capture the scanbodies. Both mastercasts with scanbodies were also scanned by a coordinate machine (Atos, GOM, Braunschweig, Germany) to create the virtual reference model. Both casts were scanned twice to determine the mean precision between the two repeated scans. The mean precision as measured using the best fit method and calculating the 3D deviation (Geomagic Control X, 3D Systems, Rock Hill, SC, USA) was 5 µm. The first scan of each mastercast was randomly chosen to serve as the virtual reference.

After all digital impressions were taken, 10 analogue impressions (A) were made of each cast using the non-splinted pick-up technique with a polyether impression material (Impregum Penta, 3M Oral Care, St. Paul, MN, USA). The implant and abutment analogues were connected to the impression copings and 20 impressions were poured out in type IV gypsum (Golden brown Fujirock EP Classic, GC, Leuven, Belgium). Then, the gypsum casts were individually scanned using a lab-scanner (IScan D104i, Imetric, Courgenay, Switzerland) with a maximum resolution of 5 µm to create a virtual cast, which reflects the workflow used in the dental lab.

The virtual implant/abutment models were created by superimposing the scan of the scanbody with the CAD-image of the scanbody and corresponding analogue using Geomagic Qualify (3D Systems, Rock Hill, SC, USA). The six analogues and scanbodies were saved as a separate STL file to be used for the measurements.

This process was repeated for each of the 80 digital impressions and the 20 scans of the gypsum casts.

Afterward, both reference scans were superimposed using a best fit algorithm (Geomagic Control X, 3D Systems, Rock Hill, SC, USA) to their corresponding test scans, in order to determine the trueness. The difference in angulation and coronal linear deviation of the center of the neck of the implant was measured on all six implants for each impression. The coronal deviation was calculated as the root mean square (RMS) between the center of the neck of the reference implant and the center of the neck of the test implant. The angular deviation was defined by the angle between the axis of the reference implant and the test implant, as defined but the centers of the implant connection and implant apex (Figure 1).

Next, precision was determined by superimposing and comparing the test scans and measuring the coronal and angular deviation for each implant. This provides information on the reproducibility of each of the different impression techniques or systems. The full workflow is depicted in Figure 2.

All statistical analyses were conducted in SPSS 27, with the level of significance set at *p* < 0.05. For descriptive analyses, the median, IQR, minimum, and maximum were calculated. Kruskal–Wallis and Mann–Whitney U-test were used to evaluate the outcomes.

*p*-values were adjusted for multiple testing with the Bonferroni correction. Statistics were performed using all measurements and not only the average distance per impression. Power calculation resulted in 88% power with a sample size of 10 scans per group.

## 3. Results

### 3.1. Trueness

The trueness of the implant-level impressions is depicted in Table 1 and Figure 3. For the angular measurements, PS52 demonstrated a lower deviation compared to all other types of impressions (*p* < 0.001) and group A missed significance with PS52 (*p* = 0.072) and T4 (*p* = 0.055). All other deviations were not significant.

For the coronal measurements, PS52 performed significantly better compared to all other impressions (*p* < 0.001), except for PS51 (*p* = 1.000). PS51 was significantly better compared to T4 (*p* = 0.009) and no significance with T3 was observed (*p* = 0.080). All other deviations were not significant.

The trueness of the abutment level impressions is depicted in Table 1 and Figure 3. The angular deviation for PS52 was significantly lower compared to all other groups (*p* < 0.050). All other angular deviations were not statistically significant.

The coronal deviation of PS52 was significantly lower (*p* < 0.001) compared to all other impressions, except for A (*p* = 1.000). Group A also demonstrated a significantly lower coronal deviation compared to PS51, T3, and T4 (*p* < 0.001).

When comparing the impressions on implant or abutment level, only PS52 demonstrated a significant lower deviation in angulation on abutment level (*p* = 0.020). A significantly lower coronal deviation on the abutment level was observed for the PS52 and A impressions (*p* < 0.001) and T3 just missing significance (*p* = 0.050).

### 3.2. Precision

The precision of the implant-level impressions is depicted in Table 1 and Figure 3.

For the angular deviation, PS52 was significantly lower compared to all other impressions (*p* < 0.001). Group A and PS51 were not statistically different (*p* = 1.000) and demonstrated a significant lower angular deviation (*p* < 0.001) compared to T3 and T4 (*p* = 1.000).

For the coronal deviation, PS52 was significantly lower compared to all other impressions (*p* < 0.001). Group A and PS51 were not statistically different (*p* = 0.273) and demonstrated a significantly lower angular deviation (*p* < 0.001) compared to T3 and T4 (*p* = 1.000).

The precision of the abutment level impressions is depicted in Table 1 and Figure 3.

For the angular measurements, PS52 demonstrated a significantly lower deviation compared to all other types of impressions (*p* < 0.001). All other deviations were not statistically significant, except for T3–T4 (*p* < 0.001).

There was no significant difference in terms of coronal deviation between PS52 and A (*p* = 1.000), but both demonstrated significantly lower deviations compared to the other impressions (*p* < 0.001). All other coronal deviations were not statistically significant (*p* = 1.000).

When comparing the impressions on implant or abutment level, no significant difference was found in the angular or coronal deviations for PS52 (*p* = 0.998 and *p* = 0.410) and the coronal deviation of T4 (*p* = 0.242). All other measurements were statistically significant (*p* < 0.05).

## 4. Discussion

The first null hypothesis can be rejected for trueness, since the conventional impression method performed significantly better compared to the digital impressions in terms of the coronal deviation, except for Primescan v5.2. Overall, Primescan v5.2 demonstrated the lowest discrepancies in trueness and precision and performed as good as the analogue impression in terms of coronal deviation and even better in terms of angular deviation.

Compared to other studies, Amine et al. [19] concluded that conventional implant impressions in the edentulous mandible, taken on the implant level, was inferior compared to the digital impression. In contrast, Huang et al. [16] reported that conventional impressions, taken on abutment level, were superior compared to intra-oral scanning. This is similar to the findings in our study. On the abutment level, Primescan v5.2 could achieve the same level of accuracy as the conventional impressions, but outclassed the conventional impression on implant level. Additionally, no significant difference could be found between the conventional impression and the other digital impression systems for trueness on the implant level model.

The second hypothesis can only be rejected for Primescan v5.2 and the analogue impression. The coronal and angular trueness of Primescan v5.2 and the coronal trueness of the analogue impression were significantly better on the abutment level compared to the implant level impression. Although not statistically significant, all other impressions also demonstrated a tendency toward a better trueness on abutment level.

Since an internal conical connection has a tighter fit and more friction with the implant compared to an external implant or abutment connection, it is plausible that deformation of the analogue impression occurs when it is removed [20,21]. This is especially the case when implants are tilted [15]. For the digital impression, Alikhasi et al. [20] suggested that there was no difference between internal and external connections in terms of accuracy. This was also confirmed in our study, where only Primescan v5.2 demonstrated an even better result on the abutment model.

Overall, most of the abutment-level impressions were more accurate compared to the implant-level impressions. The use of additional components such as abutments will increase the number of connections and thereby the risk of errors. However, this was not the case in our study, since the abutment model demonstrated a better trueness, especially at the coronal level. This could be caused by the nature of the connection, where the tightening force may play a role [7]. In contrast to an external connection, where a flat-to-flat interface is achieved, the internal conical connection will demonstrate more axial discrepancy when connecting the scanbody, depending on the insertion torque [7,22].

The higher the torque, the deeper the scanbody will be inserted in the implant. All scanbodies (abutment and implant level) were hand tightened, without the use of a torque controller, which may have led to a slight variation in the final vertical position of the scanbody in the implant. This could also explain the better values for the abutment model, where a vertical stop prevents the scanbody from sinking deeper in the abutment.

A misfit smaller than 150 µm can be considered as clinically acceptable since it will not induce biological or technical complications over time [9]. Based on our findings, only Primescan v5.2 was able to produce consistent scans within this threshold. Even the analogue impression on implant level surpassed the 150 µm deviation threshold a couple of times, although the analogue abutment impression demonstrated the lowest outliers. More recent studies put the required level of fit at 30–50 µm, since implants may move up to 50 µm within the bone [10,23]. Only Primescan v5.2 and the analogue impressions on abutment level were sufficiently accurate to produce tension free restorations. However, Huang et al. reported 100 µm as a maximum acceptable misfit, which demonstrated the lack of consensus about the true clinical implication of the coronal deviation.

In terms of the angular deviation, 0.4° has been considered acceptable in several studies [15,16]. Revell et al. [18] stated that all angular deviations under 0.75 degrees should have no negative clinical impact. In our study, PS52 demonstrated the lowest median angular deviation (0.18°–0.32°), while Trios 3 and Trios 4 had similar angular deviations with their medians between 0.35° and 0.66°. The angular deviation of the conventional implant impression remained below 0.4°.

Concerning precision, the analogue impression showed a high number of outliers. This is partially caused by discrepancies in one implant level impression. Splinting the impression copings could have reduced deformation of the impression and improved the outcome. However, overall, outliers of precision were found for all impression systems and techniques. This demonstrates that making digital or conventional impressions for full arch implant rehabilitation is still difficult and hard to reproduce.

Mizumoto et al. [1] concluded that post processing may also have a significant effect on the accuracy of IO scans. The software needs to stitch the images accurately and filter noise such as parts of the hands, lips, tongue, and suction device, which were scanned unintentionally. In this study, two versions of the Cerec SW software for Primescan were used. While software version 5.1 used an algorithm that automatically fills in small holes in the scan, the updated version 5.2 did not. This resulted in a more reliable scan and improved the level of accuracy. Additionally, less outliers were detected, which means that more scans will achieve the required level of trueness to be used for fabrication of an implant-supported prosthesis.

Some studies that have evaluated the accuracy of full-arch digital implant impressions reported lower discrepancies compared to our findings. Cakmak et al. [11] reported a trueness of 60 µm for Trios 3, while Vandeweghe et al. [13] found 28 µm deviation for Trios 2. However, these studies did not evaluate the position of the implants, but of the scanbodies. This does not represent the clinical and laboratory setting and can result in unreliable results, since it does not include the connecting procedures and errors. Pan et al. [22,24] investigated the distortion of the CAD workflow and reported that 30 µm distortion can be expected. This also includes errors generated during the data acquisition. An additional error is caused by the tolerance of the implant components, which varies among brands and depends on the attachment procedure and repeated repositioning of the scanbodies [22]. Therefore, all measurements were performed using the CAD file of the implant analogues in our study.

Another benefit of this technique is the replacement of the original mesh (Pointcloud), acquired from 3D acquisition during scanning, with the corresponding scanbody and implant from the manufacturer’s library. This allows for a more reliable comparison when using the best fit method for superimposing, as all test files have exactly the same points and geometry and only reliable points were used for the best fit algorithm [25]. Another study, using the best fit method, compared matching the CAD files and the mesh files, the group with the CAD files showed consistent lower results [8].

The reference scanner (Atos, GOM, Braunschweig, Germany) used to perform all trueness measurements has an accuracy of 1–2 µm. In most of the literature, a conventional optical model scanner has been used as a reference, which has a precision of 6–10 µm [21]. This has been seen as a limitation in comparison to the CMM system, which has an accuracy of 1 µm [9,19,21].

To superimpose the different virtual images, a ‘best fit’ algorithm was applied. This technique may cause errors during the process, which has an influence on the accuracy assessment [16]. If one of the six implants demonstrates a major displacement, the error will be distributed over the other five implants, thereby underestimating the error on the implant with the major displacement.

Because this is an in vitro study, no interfering tissues (cheeks, tongue, …) were present to create noise and interrupt the scanning protocol, which can lead to false image stitching [14,15]. Knechtle et al. [15] showed that with increasing mobility of the gums, the scanning process became less accurate.

Intra-oral scanning is also more difficult due to space limitations and a different optical behavior of the tissues when hit by light compared to extra-oral scanning of a gypsum cast [8].

## 5. Conclusions

Digital impressions for full-arch implant supported prostheses can be as accurate as conventional impressions, depending on the intra-oral scanner and software. In vivo research is necessary to confirm these results. Overall, abutment level impressions were more accurate compared to implant level impression because of the flat to flat connection, which provides a vertical stop.

## Figures and Tables

**Figure 1 jcm-11-00594-f001:**
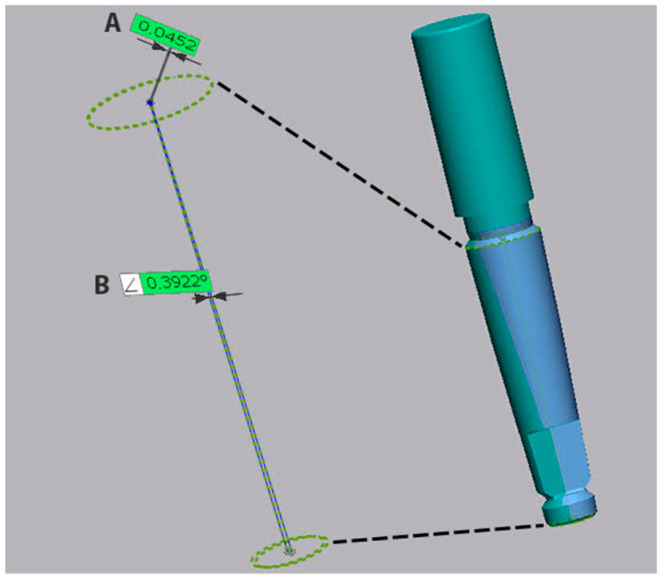
Angular (**A**) and coronal (**B**) measurements on one of the implants in the impression.

**Figure 2 jcm-11-00594-f002:**
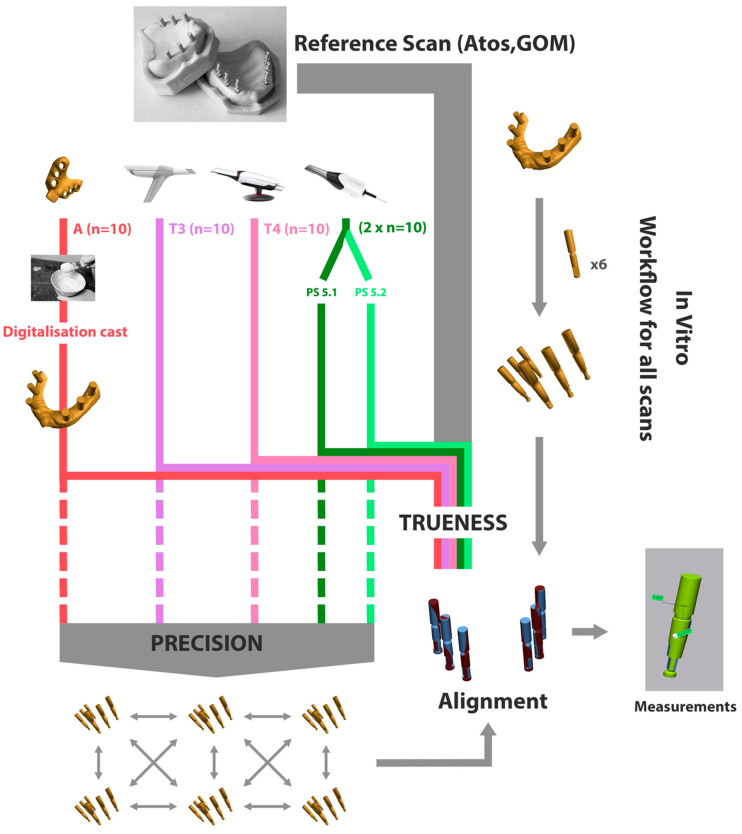
Workflow for trueness and precision analyses. The same workflow was followed for both models resulting in 40 digital scans and 10 conventional impressions for each model. Each of the 50 test scans was aligned with their corresponding reference scan (Atos, GOM, Braunschweig, Germany) for the trueness comparison and each group of ten testscans was aligned with each other for a precision comparison. After alignment, coronal and angular deviations were calculated for each of the six implants.

**Figure 3 jcm-11-00594-f003:**
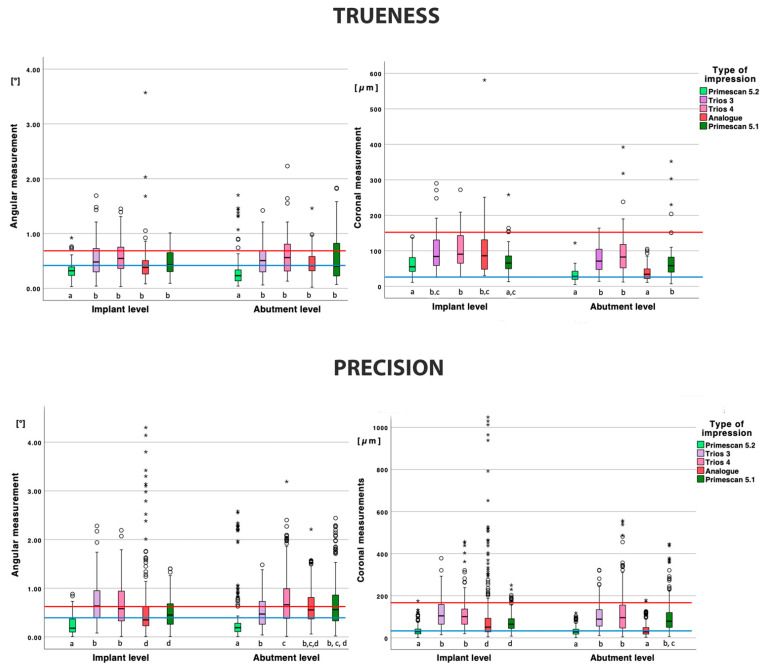
Boxplot for angular and coronal measurements for all types of impressions and for both connections. Boxplots labeled with the same letter (a,b c or d) in each of the eightgroups separately were not statistically significant (*p* > 0.05). Red line: 0.75° or 150 µm; Blue line: 0.4° or 30 µm.

**Table 1 jcm-11-00594-t001:** All descriptives for trueness and precision for both types of connection with *p*-values for the Mann–Whitney U-tests.

		Angular Measurements [°]									Coronal Measurements [µm]								
		Implant Level				Abutment Level					Implant Level				Abutment Level				
		Median	IQR	Min	Max	Median	IQR	Min	Max	*p*-Value	Median	IQR	Min	Max	Median	IQR	Min	Max	*p*-Value
TRUENESS	PS52	0.32	0.16	0.03	0.92	0.23	0.21	0.04	1.70	0.020	55	39	11	140	28	24	5	122	<0.001
	T3	0.48	0.44	0.04	1.69	0.51	0.39	0.06	1.42	0.885	84	73	25	290	71	59	14	164	0.050
	T4	0.54	0.40	0.03	1.45	0.56	0.51	0.13	2.23	0.921	91	80	27	272	83	67	12	392	0.116
	A	0.38	0.25	0.08	3.57	0.41	0.28	0.02	1.46	0.434	86	84	30	581	34	27	11	105	<0.001
	PS51	0.43	0.36	0.09	1.01	0.40	0.62	0.07	1.83	0.950	66	37	13	258	58	44	7	352	0.216
PRECISION	PS52	0.18	0.27	0.01	0.88	0.19	0.17	0.01	2.58	0.998	29	25	2	175	28	23	1	118	0.410
	T3	0.64	0.55	0.08	2.28	0.47	0.47	0.04	1.48	<0.001	105	94	15	378	89	78	11	322	0.004
	T4	0.58	0.61	0.01	2.19	0.66	0.61	0.01	3.19	0.048	101	72	20	457	96	109	5	557	0.242
	A	0.35	0.39	0.01	4.30	0.56	0.44	0.06	2.21	<0.001	51	64	5	1048	29	31	3	179	<0.001
	PS51	0.45	0.42	0.01	1.40	0.56	0.54	0.02	2.44	<0.001	65	47	8	250	79	71	7	445	<0.001

## Data Availability

Data are available from the corresponding author upon request.

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
