# Peer review of "In Vitro Accuracy of Digital and Conventional Impressions for Full-Arch Implant-Supported Prostheses"

_jcm, 2022, doi:10.3390/jcm11030594_

Round 1

Reviewer 1 Report

This study presents a comparison between conventional and digital scanning for full-arch implant-supported prostheses. Two questions are raised : the influence of the level of the scan (implant level or abutment level) and the comparison between 4 IOS / software combinations.

The question is of major interest at the moment and is well conducted. Precisions and modifications will allow for an even more interesting article.

Abstract

  • l14 : Which is the reference scan ?
  • Coronal measurement is not a frequent denomination. Please explain.

Introduction

  • Reference to orthodontics seems irrelevant considering the numerous data related to prosthodontics.
  • “This variation might be caused by operator experience, scanning conditions and 50 strategy, different types of scan bodies and the type of implant connection [11]”. The type of implant connection isn’t usually considered as an important parameter for accuracy. Please revise.
  • Problematics: Type of connection. It seems that the authors understand “implant vs abutment” impression whereas one could understand internal vs external connection. Please precise.

Materials and methods

  • How did the authors verify that implants were placed exactly in the same place in the two models ? Please explain.
  • How were situated the implants in relation to each other ? On the figure, we understand that they are quite parallel. This is really important given that physical impressions were taken and we know that angulation is a parameter for precision in this case.
  • How was precision measured in the reference scans (I understand the coordinate machine) ? The authors say that they used Geomagic software. Did they perform a best fit ? Please, use another expression that “reference scans” for this reference measurement.
  • Why were the models scanned without the scan bodies at first ? Did the authors use these scans or did they perform a two-steps acquisition (model, then model + scan bodies) ? Was that useful ?
  • Trueness and precision were measured with a best-fit super-imposition process. Was it perform separately for each scan body or with the six scan bodies of the same scan ?
  • Outcome measurements are not sufficiently described. The authors say that they used the center of the neck of the implant as the reference point. With one point, how did they calculate an angulation ? Please introduce here the coronal trueness measurement that you announced in the abstract.
  • Statistical analysis : How were the statistics performed ? Implant per implant or impression per impression ? Were equality of variances verified each time ? Please precise.

Results

  • A had and a near missed significance … cut and paste error ? Not a very scientific expression !
  • What is A and group A ? they were not introduced in the materials and methods section.
  • Fig 2, legend. The significance presentation is unclear. Please choose another way to present these results.

Discussion

  • Once again, the reader really needs an explanation about the coronal deviation.
  • « Overall, most of the abutment-level impressions were more accurate compared to the 200 implant-level impressions. The use of additional components, such as abutments, will 201 increase the number of connections and thereby the risk for errors.

However, this was not the case in our study, since the abutment model demonstrated 203 a better …” : please regroup in the same paragraph. As presented, it is misleading.

  • Angulation was demonstrated as the major issue now in complete-restoration digital scanning, as distance discrepancies seem limited with PrimeScan and Trios. Please insist on your findings ! Beginning with better explaining angulation measurement….
  • “However, these studies did not evaluate the position of the implants, but of the scan bodies.” Please explain why and how you succeeded in evaluating the position of the implants.
  • Indeed, best-fit is not recognized as the best way to perform trueness and precision measurement. Please explain your choice.

Conclusion

Showing that abutment level scans are truer and more precise that implant level scans seems the major result of this study. Maybe repeat here why.

Author Response

Dear reviewer,

Thank you so much for taking the time to review our manuscript.

In the attachment you can find our respons.

Reviewer 2 Report

line 66

at the end of the introduction part, “the purpose” was missing. I recommend addressing the purpose of the study in front of the first null hypothesis

line 78
the master casts were scanned twice, with the first one without scanbodies and the second one with scanbodies. These procedures may improve the accuracy of scan procedures. However, are these procedures routine in clinical situations? 

line 92
splinting impression copings was not performed for the conventional (analogue) impression. This definitely influenced the final accuracy of the analogue impression. Many studies reported inconsistent results for internal connection implants. However, for external implants (in this study, uniabutments were used to represent flat-top connection) splinting impression copings produce more accurate results compared to non-splint

line 93
gypsum casts fabricated by analogue impression was scanned by lab scanner to produce digital models. However, this procedure definitely produce an additional error. Reference 7 used digital inspection for scan files (models) and analogue CMM for analogue model not to produce unnecessary distortion for the analogue impression.

probably, Kruskal-Wallis for comparison between different impressions, and M-H U test for implant and abutment. Even figure 2 showed the results as graphs, the P values should be addressed in table 1. Table 1 only represented the p values for implant and abutment comparison (hypothesis 2) not for the comparison between impression differences (PS52, T3, T4, A, PS51) please put the p value for the K-W test and Bonferroni connection 

Author Response

(The authors gave the same response as above.)

Reviewer 3 Report

Dear Editor,

Regarding the submitted manuscript “  Accuracy of digital and conventional impressions for full-arch  implant-supported prostheses.” this review will report a detailed appreciation.

The presented study is intended to be an in vitro study to evaluate accuracy of full-arch impressions.

The article is well written, and the proposed objectives were attained. It is this referee belief that some changes are needed before considering the manuscript for publication:

  • Some misspellings were found through the manuscript and should be corrected
  • Title should be revised since it should mention that it is an in vitro study.
  • Introduction:
    1. I advise to use the definition of trueness and precision of the ISO 5725-1 and state in the M&M how they were assessed for the proposed study (rms? Linear measures? SD?...)

  • When looking into the material and methods section some corrections and clarifications are needed:
    1. The statistical sample size determination and power analysis needs to be stated for the evaluated variables.
    2. The authors mention that implants were placed in a 3d printed resin maxilla. How was the stability of the implants ensured in the printed maxilla and were any of the implants tilted? Was a guided stent used to ensure that implants were placed in similar angulations and places between the 2 models? These considerations need to be stated.
    3. Being that the printed model was without soft silicone to reproduce the soft tissues how the authors ensured that the torqued scanbodies and abutments did not suffer resin interference while being tightened?
    4. The authors need to mention the impression materials and gypsum brands and batch numbers and if the conventional impressions were splinted or not
    5. Why only 2 digitization’s were performed with the reference scanner? Considering manufacturer’s reported scanner accuracy, the authors should perform at least 3 readings per guide to assess scanner bias variability. Additionally, they need to state after the scanners which was used as the reference model and why
    6. The authors should state that the scanners were calibrated according to manufacturer’s instructions before use.
    7. Definition of trueness and precision should be
    8. The authors state that “The virtual implant/abutment models were created by superimposing the scan of the scanbody with the CAD-image of the scanbody and corresponding analogue using Geomagic Qualify (3D Systems, SC US)”. Considering that CAD-Image of the scanbody with corresponding image cannot be exported from the cad software to geomagic, the authors need to state how this 3D image was obtained since outcome determination is dependent on the best fit superimposition of these images.
    9. The authors mention “The difference in angulation and 3D deviation of the center of the neck of the implant were measured for each analogue”, it needs to be a clear description how the angulation and 3D deviation was performed. When the authors mention 3D deviation measurements are they linear distances or overall root mean square results? Regarding angulation how it was measured, and between what implants in the model? More information in this part of the methods is needed.
    10. In figure 1 for standardization purposes the study design, the number of samples and other details need to be clearer. Additionally, the image used to show the measurements performed needs to be described
    11. When performing the best fit model was overall or one of the implants considered as reference?
  • Results:
    1. The description of the results needs reformulation and I advise the support of a professional statistician since I believe that the manuscript will gain by a more detailed analysis.
  • Discussion:
    1. The discussion is a bit to extensive and careful should be addressed since extrapolations should not be performed between in vitro results to in vivo.

The study design and Material and methods section need some changes before being considered for  publication.

Author Response

(The authors gave the same response as above.)
